# Exploring Toxin Genes of Myanmar Russell’s Viper, *Daboia siamensis*, through De Novo Venom Gland Transcriptomics

**DOI:** 10.3390/toxins15050309

**Published:** 2023-04-26

**Authors:** Khin Than Yee, Jason Macrander, Olga Vasieva, Ponlapat Rojnuckarin

**Affiliations:** 1Department of Medical Research, Ministry of Health, Yangon 11191, Myanmar; 2Department of Biology, Florida Southern College, Lakeland, FL 33801, USA; 3Institute of Integrative Biology, University of Liverpool, Liverpool L69 7ZB, UK; 4BioSynthetic Machines, Inc., Chicago, IL 60062, USA; 5Excellence Center in Translational Hematology, Division of Hematology, Faculty of Medicine, Chulalongkorn University, Bangkok 10330, Thailand

**Keywords:** venom gland transcriptomes, Russell’s viper, *Daboia siamensis*, next-generation sequencing, toxin genes, Venomix pipeline

## Abstract

The Russell’s viper (*Daboia siamensis*) is a medically important venomous snake in Myanmar. Next-generation sequencing (NGS) shows potential to investigate the venom complexity, giving deeper insights into snakebite pathogenesis and possible drug discoveries. mRNA from venom gland tissue was extracted and sequenced on the Illumina HiSeq platform and de novo assembled by Trinity. The candidate toxin genes were identified via the Venomix pipeline. Protein sequences of identified toxin candidates were compared with the previously described venom proteins using Clustal Omega to assess the positional homology among candidates. Candidate venom transcripts were classified into 23 toxin gene families including 53 unique full-length transcripts. C-type lectins (CTLs) were the most highly expressed, followed by Kunitz-type serine protease inhibitors, disintegrins and Bradykinin potentiating peptide/C-type natriuretic peptide (BPP-CNP) precursors. Phospholipase A_2_, snake venom serine proteases, metalloproteinases, vascular endothelial growth factors, L-amino acid oxidases and cysteine-rich secretory proteins were under-represented within the transcriptomes. Several isoforms of transcripts which had not been previously reported in this species were discovered and described. Myanmar Russell’s viper venom glands displayed unique sex-specific transcriptome profiles which were correlated with clinical manifestation of envenoming. Our results show that NGS is a useful tool to comprehensively examine understudied venomous snakes.

## 1. Introduction

Russell’s viper is a widespread species found in South and Southeast Asia, as well as China. Based on previous molecular [1] and morphological [2] studies, the *Daboia* genus has been divided into two well-defined taxa: *D. russelii* (west of the Bengal Bay) and *D. siamensis* (east of the Bengal Bay). The species designation and regional distribution align with variations observed in the clinical features of Russell’s viper envenomation [2,3]. The venom proteomes of Russell’s viper vary across their geographical distribution and have been attributed to these differences [4,5,6,7,8,9].

Snake venoms are mixtures of proteins working synergistically to produce various toxic functions. These compounds evolved to immobilize, kill and initially digest their preys [10,11]. The myriad toxin compositions and functions are a byproduct of their interrelation between natural selection and biochemical processes shaping the protein–protein or protein–cellular interactions. Venom variation is found not only across lineages, but also within individuals of the same species [12]. Additional factors influencing venom variation include age, sex, diet, season, frequency of extraction and a post-extraction process [13,14,15]. The ontogenetic change and venom gland location can also influence the venom composition in an individual snake [16].

Transcriptomic and proteomic approaches have been used to better understand venom diversity [17,18,19]. Transcriptomic studies using next-generation sequencing (NGS) techniques deliver a much broader view of the venom protein coding genes and a deeper understanding of toxin transcript abundance. Transcriptomic approaches permit the identification of unique isoforms that are candidates for drug discovery [20]. Using these techniques, multiple toxin gene copies have been identified within comprehensive datasets, aiding our understanding of how venom protein gene families have evolved. Each toxin family shows a unique evolutionary rate, which can be linked to a change in ecological opportunities, such as predation [21].

Proteomic approaches are often used to provide an expanded overview of venom composition and expression. Previous proteomic studies of Russell’s viper from different countries have shown that toxin gene families exhibit varying venom compositions throughout their distribution [4,6,7,8,22,23].

Despite the differences across their population observed in proteomic data, the Russell’s viper protein-coding toxin genes have not been previously investigated through a comprehensive transcriptomic analysis, nor have any Russell’s viper studies focused on sex-specific venom expression despite anecdotal evidence indicating size- or sex-specific envenomation symptoms in humans. The present study aims to generate a sex-specific de novo venom-gland transcriptomes of Myanmar Russell’s viper (*D. siamensis*) and provide a deeper insight into the correlation between the toxin composition and pathophysiology of Russell’s viper envenomation. The results of the current study can be used to validate diverse predicted toxins as well as identify novel toxin families in this species that are yet to be identified.

## 2. Results

### 2.1. De Novo Transcriptome Assembly

Sequencing of the cDNA libraries yielded a total of 78.64 and 74.44 million clean reads for male and female Myanmar Russell’s vipers, respectively (Table 1). De novo assembly using the Trinity [24] program created 88,325 contigs (N50 = 1870) comprising 77,668 Unigenes (N50 = 1397) for the male snakes and 50,858 contigs (N50 = 1379) comprising 46,884 Unigenes (N50 = 1183) for the female snakes.

### 2.2. Categorization of Transcripts and Gene Expression

The expression levels of toxin transcripts from both samples were compared with those of non-toxin transcripts from tissues of the same species (Bioproject PRJNA269317, Biosamples SAMN03081270) (Appendix A).

The toxin transcripts were confirmed to be highly expressed in venom gland tissues. They were categorized according to TPM values. The individual toxin transcripts with a TPM value of more than 100 were categorized as highly expressed, a TPM value of 10 to 100 as moderately expressed and a TPM value of less than 10 as lowly expressed toxin groups. Only highly expressed groups were considered for further detailed evaluation and description in this study. Non-toxin full-length transcripts, such as glutaminyl-peptide cyclotransferase (highly expressed), tomoregulin-2 (moderately expressed) and endothelin-converting enzyme (lowly expressed), were also identified in the current transcriptome. Highly expressed transcripts were no less than 150-fold higher in the venom gland when compared to samples comprising other tissues (brain, kidney, heart, lung, spleen and liver).

The highly expressed venom transcripts were categorized into 23 toxin gene families, including 53 unique full-length sequences. The most prevalent transcripts were C-type lectins (CTLs) (26%), Kunitz-type protease inhibitors (KSPIs) (19%), disintegrins (18%) and bradykinin potentiating peptide/C-type natriuretic precursors (BPP-CNPs) (16%). The remaining 21% consisted of phospholipase A_2_ (PLA_2_), serine proteases, metalloproteinases, vascular endothelial growth factor (VEGF), L-amino acid oxidase (LAAO) and cysteine-rich secretory proteins (CRISPs) (Figure 1).

When comparing males and females as shown in Figure 2, the male sample had CTLs, KSPIs, BPP-CNPs and disintegrins as the most highly expressed toxin groups, while metalloproteinases, CTLs, BPP-CNPs and KSPIs were the top highly expressed toxin groups in female snakes. The expression levels of transcripts for most male toxin groups were higher than those in the female-mapped reads, with 1.39- to 11.68-fold differences. However, the number of transcripts for serine proteases, glutaminyl-peptide cyclotransferase-1 and metalloproteinases, were 1.16, 1.22 and 2.33 times higher in female than male venom glands, respectively. Some toxin transcripts, such as cathelicidin-related protein (Vipericidin) and techylectin-like protein (Veficolin), were only identified in male samples.

### 2.3. Snake Venom C-Type Lectins (CTLs) and Related Proteins

The present study revealed 23 CTL transcripts (4 full-length and 19 partial-length transcripts), including 7 different subtypes of CTLs. They are (I) newly identified *Bitis gabonica* C-type lectin 1 homolog, (II) *D. siamensis* p31 β subunit, (III) RVV-X LC1, (IV) *D. siamensis* p68α subunit and *Microvipera lebetina* Snaclec A15 homolog, (V) Dabocetin α and *Microvipera lebetina* Snaclec A12 homolog, (VI) RVV-X LC2 and (VII) *D. siamensis* p68 β subunits and *D. siamensis* Snaclec-7 (Appendix A). Figure 3 shows the gene tree of these sequences. RVV-X CL1 and RVV-X CL2 linked to the metalloproteinase (RVV-X) to act on factor X of the coagulation cascade.

Our sequence alignment (Appendix A) identified that these sequences contained six conserved cysteine residues (typically eight in total for each sequence), along with a WIGL conserved motif for CTL and the WND Gla-domain binding site.

### 2.4. Kunitz-Type Serine Protease Inhibitors

A total of 15 Kunitz-type serine protease inhibitor (KSPI) transcripts were identified, including seven full-length and eight partial-length transcripts. Thirteen unique isoforms of KSPIs were observed (Table 2).

### 2.5. RTS-Disintegrins

A total of three full-length disintegrin transcripts were identified, including two isoforms of RTS-containing disintegrins which were also described as Dis1a and Dis1b in our previous paper [25]. In the gene tree analysis, Dis1a shared a high sequence similarity with jerdostatin from *Protobothrops jerdonii* (Q7ZZM2), while Dis1b was in the same branch as acostatin-α from *D. siamensis* (AUF416558) (Appendix A). In the sequence alignment, the N-terminal sequence of Dis1b was more similar to that of ascostatin-α than jerdostatin and it possessed an RTS motif instead of an RGD motif (Figure 4).

### 2.6. Bradykinin-Potentiating Peptides and C-Type Natriuteric Peptide (BPP-CNP) Precursors

A total of four full-length BPP-CNP transcripts were identified in this study. In Figure 5, the repeated BPP units were present in a BPP-CNP precursor. These BPP units appeared in tripeptide metalloproteinase inhibitors. CNPs were found to be variable between these sequences. The sequence alignments of the natriuretic peptide domains of Myanmar Russell’s viper BPP-CNP precursors with those of human and other snakes are shown in Figure 6.

### 2.7. Phospholipase A_2_ (PLA_2_)

In the focal transcriptomes, there were five full-length and two partial-length PLA_2_ candidates. Five full-length transcripts contain three group II PLA_2_ and two group III PLA_2_ transcripts. Three of the group II PLA_2_s (acidic and basic PLA_2_) were highly expressed (TPM >10,000–14,000), followed by two group III PLA_2_ transcripts (TPM 14–35), whereas two partial-length transcripts, group IIF PLA_2_ homologs of *Crotralus atrox* and a crotoxin homolog from *Protobothrops guttatus*, were the lowest expressed transcripts (TPM 1–8). The highly expressed group II PLA_2_ transcripts were aligned with PLA_2_ from Eastern Russell’s viper PLA_2_s, DsM b1and daboiatoxin A from Myanmar, PLA_2_-I from China and RV-7 from Taiwan (Figure 7).

### 2.8. Snake Venom Serine Proteases (SVSPs)

There were 14 transcripts (2 full-length and 12 partial-length) annotated as serine proteinases that were highly expressed (TPM 448.14–5495.65). Four transcripts gave an annotation as RVV-V_γ_ of *D. siamensis* (Appendix A), three transcripts as α-fibrinogenases and one transcript as β-fibrinogenase of *D. siamensis*. Four transcripts were homologous to VLSP-1 (Figure 8) and one transcript to VLSP-3 from *Macrovipera lebetina*. The remaining one was partially similar to VSPBF-DSABSI from *D. siamensis* and partially to VLSP-3 from *Macrovipera lebetina*. We could not find out the exact similarity due to the partial-length nature of the remaining transcripts (Table 3).

### 2.9. Snake Venom Metalloproteinases (SVMPs)

In the current transcriptomes, two full-length transcripts for RVV-X heavy chain (Russell’s viper factor X activator, A0A2H4Z2W1) and two partial-length DSAIP (*Daboia siamensis* VLAIP-like SVMP, A0A2H4Z2Y9) are the most highly expressed SVMP transcripts (>1.0–1.8 × 10^4^ TPM). Of the remaining three full-length transcripts with TPM values of 15–47, c15329_g2_i1_M_FL and c10072_g2_i1_F_FL were homologous to the A disintegrin and metalloproteinase domain (ADAM)-containing protein 10 from *Vipera anatolica senliki* (A0A6G5ZVR9) (Figure 9), and c19271_g2_i1_M_FL was homologous to SVMP from *Agkistrodon contortrix contortrix* (A0A1W7RJN7) (Figure 10). All conceptually translated proteins contained the signal peptide, a metalloproteinase domain containing a Zn^+^-binding catalytic site and a disintegrin domain. There was an additional ADAM17 MPD domain containing a CxxC motif in the c19271_g2_i1_M_FL transcript.

### 2.10. Snake Venom Vascular Endothelial Growth Factors (svVEGFs)

In the current study, seven full-length VEGF transcripts were identified including three isoforms which comprised 136, 148 and 192 amino acids. Two transcripts were homologous to VR-1 (*D. russelii*) (P67861), as shown in Appendix A. Two transcripts were related to VEGF1 (*Vipera anatolica senliki*) (QHR82857) and VEGFA-VIPAA (*Vipera ammodytes ammodytes*) (C0K3N5), as shown in Figure 11A,B.

### 2.11. L-Amino Acid Oxidases (LAAOs)

In this study, two full-length and one partial-length LAAO transcripts with 99.93% identity (84% query coverage, E value 0.0) to DrLAO (G8XQX1) were identified. The sequence alignment is shown in Appendix A.

### 2.12. Cysteine-Rich Venom Proteins (CRISPs)

There were two full-length highly expressed cysteine-rich secretory protein (CRISP) transcripts identified in the current transcriptomes which gave the annotation of Dr-CRPK from *D. russelii* (ACE73567), currently termed *D. siamensis* (Appendix A). These deduced amino acid sequences were also identical to those of CRISPs (MRV CRISP) (JZ980958, JZ980965 and JZ980970), which were expressed in previous cDNA libraries [26].

## 3. Discussion

The de novo venom gland transcriptomic analysis provides a unique profile of the toxin composition of Myanmar *D. siamensis* venom and its suggestive toxicogenomic diversity. Our previous study on the venom gland cDNA library revealed unique gene expression patterns [26]; however, the use of NGS technology allowed us to provide a broader examination into the variation, distribution and diversity of toxin transcripts. Using Trinity as our de novo transcriptome assembly program, we were able to assemble a robust transcriptome using high-quality (>98% Q20 for both male and female) raw reads and subsequent mapping for gene expression analysis. Our initial exploration into these transcriptomes relied on RSEM and gene expression information to identify focal toxin candidate transcripts; however, future studies would benefit from exploring additional assembly programs providing a multi-faceted approach to analyzing these transcriptomes and account for chimeric transcripts [27]. The venom proteome of *D. siamensis* comprised six toxin families, including serine proteinases, metalloproteinases, PLA_2_, LAAO, VEGF and CTLs [4]. The BPP-CNP transcripts containing SVMPIs were identified only in the transcriptomes due to the co-expression of SVMPI tripeptides and atrial natriuretic peptides in the same transcripts [28]. Only the processed products could be detected in the venom. Low-molecular-weight disintegrins (<10 kDa) were detected only in a transcriptome because a proteomic study could identify proteins larger than 10 kDa. The reason that CRISPs (27 kDa) were not found in the proteome was unclear.

We found sex-related differences in venom transcript expression in Russell’s viper, also detected in *Bothrops jararaca* [12], *Thamnophis elegans* [17] and *Mesobuthus martensii* [29]. These differences reflected the sex-based variations of the venom proteome and were closely associated with the biochemical and biological properties of different genders in the same species [14,30,31,32]. In *B. jararaca* and *B. moojeni*, the variability in the venom composition was associated with animal sizes, as females are larger than males [32,33]. This coincided with the sexual dimorphism of the venom donors used in the study, as females (average weight in ounces × total length in inches x girth in inches: 49.9 × 47.8 × 6.5) were larger than males (29.8 × 47 × 5.3) and more venom was collected from females. There is also a correlation between venom yield and the length of Myanmar Russell’s vipers, which is related to the severity of envenoming, time for coagulopathy development and the severity of local manifestations [34].

The transcripts corresponding to the toxin groups described below were found to be most abundant in both sexes and aligned with Myanmar Russell’s viper envenomation symptoms. Snake venom C-type lectins (snaclecs) are heterodimers of two closely related α and β subunits [35]. They target factor IX, factor X, prothrombin or α-thrombin (as pro- or anticoagulant) and platelet receptors, such as glycoprotein (Gp) Ibα, Gp Ia/IIa, or Gp VI (as pro- or antithrombotic) [36,37]. Snaclecs activate platelets and thrombus formation causing thrombo-inflammation during viper envenoming [38]. Dabocetin α together with its β subunit interacts with platelet GpIb receptor, inhibiting platelet aggregation [39]. *D. siamensis* Snaclec-7 is a β subunit of a GpIb-binding protein [40], and together with dabocetin α is involved in platelet inhibition. The *M. lebetina* Snaclec A12 homolog, originally found in *D. siamensis*, has a stronger sequence similarity to α than β subunits compared to other heterodimeric snaclecs [41].

Snake venom protease inhibitors primarily disrupt blood coagulation resulting in blood loss and the death of preys [42]. The largest and most widely distributed family is Kunitz-type protein inhibitors, which strongly inhibit serine proteases. They contain 58–60 amino acid residues, including three disulfide bridges [43]. Several Kunitz-type protease inhibitors from *Daboia* species have previously been identified.

The snake venom disintegrins are a group of small cysteine-rich proteins found in a variety of snake families. They are divided into seven groups according to disulfide patterns and sequence signatures [44]. Jerdostatin homologs (Dis1a and Dis1b) containing eight cysteines belong to group 7A. Most of these small disintegrins are synthesized from short-coding genes, instead of a proteolytic process from PII SVMPs like the majority of disintegrins [45]. The RTS-disintegrins are expressed from the *RPTLN* gene and have a high conserved signal peptide sequence with *SVMP* genes, suggesting they may share tissue-specific regulatory elements [46,47]. The disintegrin motif is responsible for binding specificity to integrin receptors [48]. Modulating disintegrin motifs is interesting for the development of antiangiogenesis, antimetastasis, antiproliferative and proapoptotic agents for cancer treatments. Lowly expressed VGD- and MLD-containing dimeric disintegrins observed in our previous study [25] were not evaluated in the current study as the candidate transcripts had TPM values lower than 100 and were therefore not analyzed.

The BPP-CNP transcripts usually carry a tripeptide repeat together with bradykinin-potentiating peptides (BPPs) and natriuretic peptide (NP) precursors [28,49]. The two tripeptides (pERW and pEKW) from Myanmar Russell’s viper completely inhibited the gelatinolytic effects of RVV-X and the fibrinogenolytic activities of daborhagin [28]. These viper-restricted inhibitory peptides protected the venom gland from autodigestion and prevented hydrolysis of other proteins in venom glands [50]. Their inhibitory activity towards the SVMP Zn^2+^ binding site [51] has made them popular targets for drug design in areas of neurodegenerative and cardiovascular diseases. Natriuretic peptides (NPs) in vertebrates promote salt secretion, regulating cardiovascular and renal systems. Three mammalian NPs are atrial natriuretic peptide (ANP), B-type natriuretic peptide (BNP) and C-type natriuretic peptide (CNP). ANP and BNP are released by cardiomyocytes in response to hypertension and hypervolemia, and CNP is produced by endothelial cells. The sequence alignment of various snake venom and human natriuretic peptides is shown in Figure 6, where five residues (F, D, R, I/V and L/F) are crucial for receptor binding [50]. Among MRV CNP transcripts, one had the same conserved amino acid residues, while remaining three possessed V instead of I and F instead of L residues. These variations should be explored in relation to their roles in terms of toxicity and their potential therapeutic properties. NPs have their specific niche in medical research in the treatment of heart failure and hypertension [51,52].

Phospholipases A_2_ (PLA_2_) catalyze a hydrolysis of glycerophospholipids, fatty acids and lysophospholipids and release arachidonic acids to trigger inflammation. Snake venom PLA_2_s (svPLA_2_) are small, calcium-dependent, secreted enzymes ranging from 12 to 14 kDa. The class I PLA_2_s are found in Elapidae and Hydrophiidae, while class II are in Crotalidae and class III are in lizard and bee venoms [53]. There are two major types in Russell’s vipers. PLA_2_s containing an N-terminal Asn (N) are found in species from China, Thailand, Myanmar and Pakistan, while the venoms of southern Indian and Sri Lankan species contain PLA_2_ with an N-terminal serine (S). They confer hypotensive with neurotoxic and myonecrotic properties, respectively [26]. Furthermore, Russell’s viper PLA_2_ showed nephrotoxicity in mice [54]. All of the three PLA_2_ transcripts from the current transcriptomes contained N-terminal Asparagine (N) and conserved Aspartic acid (D) at the 49 position. In all transcripts, 14 cysteine residues for seven conserved disulfide bonds indicated that they were GIIA PLA_2_s. They all possessed Thr41 which was found to be conserved in all the neurotoxic or myotoxic PLA_2_s. From Myanmar *D. siamensis* venom, DsM b1 had neurotoxic and lethal effects [55] and dabotoxin showed neurotoxic, myotoxic, cytotoxic, edema-inducing and indirect hemolytic activities [56]. 

Snake venom serine proteases (SVSPs) have two six-stranded β-barrels and a catalytic triad of His57, Asp102 and Ser159. Viper venoms contain a large variety of SVSPs, mostly as monomeric glycoproteins of 26 to 67 kDa. They act on almost all components of hemostatic and fibrinolytic systems and are responsible for the hemorrhage and shock after envenomation. Based on their biological roles, they are classified as activators of the fibrinolytic, procoagulant, anticoagulant and platelet-aggregating enzymes [57,58,59]. SVSPs from *D. russelii* are procoagulants that (i) activate factor V (RVV-V), (ii) have thrombin-like action causing defibrinogenation (Russelobin) and (iii) are αβ fibrinogenases with factor V-activation (RV-FVPα-δ). RV-FVPs showed in vivo anticoagulant effects in addition to in vitro procoagulant effects [60].

Snake venom metalloproteinases (SVMPs) are zinc endopeptidases varying from 20 to 100 kDa. They are the major components of viper venom, varying from 11% to 65% of the whole venom cocktail [61,62]. SVMPs have sequence and domain similarities to ADAM (A disintegrin and metalloproteinase) and can be grouped into three major classes (PI-PIII) according to their organization [53,63]. SVMPs are responsible for local and systemic hemorrhage, myonecrosis, the disruption of hemostasis mediated by procoagulant or anticoagulant effects or platelet aggregation, tissue necrosis and pro-inflammation [61]. These activities are mediated by metalloproteinase, disintegrin and/or disintegrin-like domains [62]. Snake venom metalloproteinases (SVMPs) isolated from Russell’s viper were RVV-X (85 kDa), Daborhagin (65 kDa), VRR-73 (73 kDa), VRH-1 (23 kDa) and RVBCMP (25 kDa). RVV-X is a well-characterized class P-IIId SVMP. It activates factor X via a single cleavage at the Arg^194^-Ile^195^ bond in factor X, which is responsible for disseminated intravascular coagulation (DIC) in snakebite patients [64]. RVV-X also binds to receptors on platelets and endothelial cells, resulting in clinical bleeding [60,65] and contributing to acute kidney injury in animal models [66]. The structure and biological activity of DSAIP still needs to be elucidated. RVV-X and daborhagin are found in both Myanmar and Indian RV species, while VRR-73, VRH-1 and RVBCMP are from only Indian RV [61]. Daborhagin-M is a P-IIIc hemorrhagic SVMP with high proteolytic activity toward fibrinogen, fibronectin and collagen [67]. These activities may be related to the severe bleeding in the patients.

A disintegrin and metalloproteinase (ADAM) family proteins in focal transcriptomes were expressed at moderate levels. Transcripts homologous to VIPAN (ADAM 10) from *V. a. senliki* and AGKCO (ADAM 17) from *A. c. contortrix* were found in Myanmar Russell’s viper. ADAM 10 played a major role in the signaling pathway and ADAM 17 in the processing of tumor necrosis factor-α and other cell-surface molecules [68]. Although ADAM 17 is considered to be the primary sheddase for TNF-α, ADAM 10 also worked as a TNF-α sheddase in certain cell types [69]. The biological functions of ADAM 10 and 17 from snakes require further investigation.

Snake venom vascular endothelial growth factors (svVEGFs) play important roles in angiogenesis (endothelial growth and proliferation), hypotension and vascular permeability by binding to VEGF receptors. svVEGFs are 25-kDa, homodimeric heparin-binding proteins with 50% identity to VEGF-A165. They have a VEGF homology domain (VHD), with eight conserved cysteines forming a cysteine knot. The C-terminal variable region corresponds to heparin and NP-1 (coreceptor) binding sites [70,71]. VR-1, a svVEGF from *D. russelii,* increased endothelial cell proliferation, vascular permeability and hypotension through the specific activation of endothelial KDR (kinase insert domain-containing receptor) [72]. The VR-1 isolated from *D. siamensis* also increased endothelial cell proliferation [73]. The markedly enhanced vascular permeability caused by svVEGFs can lead to hypotension and shock.

## 4. Conclusions

The de novo venom gland transcriptomic analysis provided a unique profile of venom gene expression in *D. siamensis* from Myanmar. The expression patterns of toxin-encoding genes showed high structural redundancy in comparison to non-toxin genes, suggesting distinct functions of the variants in each group. The expression patterns of venom genes were also sex-specific. The distribution and expression levels of the principal toxin components yielded deeper insights into the toxic syndromes of Russell’s viper envenoming, which included bleeding, hypotension and renal failure. Moreover, the full-length toxin sequences from this study would be the references for future proteomic analysis. Newly identified venom components will be a source for future structure–functional relationship studies and the development of new therapeutic modalities. Taking into account that the genomes of these species were not yet completed, this study provides rich data for a fast advancement in toxicogenomic research to study evolutionary origins and diversity in snakes.

## 5. Materials and Methods

### 5.1. Sample Preparation

Four (two male and two female) adult Russell’s viper, *D. siamensis*, were captured from Kyun Gyan Gon Township, Yangon, Myanmar. Venom milking was carried out prior to tissue harvesting to stimulate the toxin transcription. The snakes were allowed to rest for four days to maximize RNA transcription. The venom glands were then removed, cut into pieces smaller than 5 × 5 mm and stored in an RNAlater solution (Ambion, Austin, TX, USA) at 4 °C overnight before transferring to −80 °C.

### 5.2. RNA Extraction, cDNA Library Construction and Sequencing

Total RNA from venom gland tissues was extracted using Trizol reagent (Life Technologies, Carlsbad, CA, USA) or a Total RNA Purification Kit (Jena Bioscience GmbH, Jena, Germany). The sample were stored at −80 °C. Subsequently, mRNA from samples was isolated using PolyAT Tract mRNA Isolation System (Promega, Madison, WI, USA) or a FastTrack MAG mRNA Isolation Kit (Invitrogen, Carlsbad, CA, USA). Both systems used MagneSphere^®^ technology. Subsequently, mRNA was precipitated, concentrated using 3M sodium acetate and isopropanol and stored at −80 °C. Isolated mRNA was suspended in RNase-free water and sent to Macrogen Inc. Geumchun-gu, Seoul, Republic of Korea. The mRNA sequencing libraries were prepared using the TruSeq RNA sample preparation kit (Illumina Inc., San Diego, CA, USA) with selected insert sizes of 200–400 bp. Libraries were then sequenced on the Illumina HiSeq2000 platform using (2 × 100 bp) paired-end reads.

### 5.3. De Novo Transcriptome Assembly and Expression Quantification

The quality of raw data or raw reads from the Illumina platform was assessed with the quality assessment software FastQC (http://www.bioinformatics.babraham.ac.uk/projects/fastqc; accessed on 18 November 2014) and cleaned up using Trimmomatric (0.32) (http://www.usadellab.org/cms/?page=trimmomatic; accessed on 20 November 2014) to remove adaptor sequences, nucleotide errors and low-quality sequences. Two assembly tools, Trinity [24] and CLC Genomic Workbench 7.5 (CLC bio, a QIAGEN Company, Aarhus, Denmark), were used to assemble and compare alternative pathways for de novo transcriptome assemblies. The quality assessment tool QUAST was used to evaluate the quality of the assembled transcriptomes [74], indicating that the Trinity assembly had a higher quality score than the CLC genomic assembly (Appendix A). Thus, assembled data via Trinity were used for downstream analysis.

The transcriptomes were assembled in the Trinity software (r20140717) [24]. For each transcriptome, expression values were determined as transcripts per million (TPM) and fragments per kilobase million (FKPM) were calculated in the program RSEM [75,76] as part of the Trinity package. The raw data of the assembly were uploaded to NCBI database under BioProject PRJNA545823, Biosample SAMN11938797 for males and SAMN11939170 for females. The expression levels in terms of both TPM and FKPM of toxin transcripts were mapped to transcripts from non-venom gland tissues, such as brain, kidney, heart, lung, liver and spleen (Bioproject PRJNA269317, Biosamples SAMN03081270), of *D. siamensis*.

### 5.4. Venom Gland Transcript Classification

Toxin transcripts from the two transcriptomic datasets were identified and characterized using Venomix [77]. The Venomix pipeline identified candidate toxin genes based on BLAST hits (E-value 1 × 10^−20^) against the ToxProt dataset [78]. A transcript was considered a “candidate” if the transcript had a lower E-value associated with a toxin than with a non-toxin protein in Uniprot. The candidate transcripts were translated into their protein sequences using Transdecoder [79] and further evaluated in ToxClassifier [80]. A predicted protein sequence with a score of >1 was considered a toxin candidate. The annotation of generated transcript sequences was aligned by standalone BLASTP (v. 2.10.0) to the protein database UniProtKB/Swiss-Prot [81] (Swissprot, Swiss Institute of Bioinformatics, Lausanne, Switzerland) of Serpentes (taxid: 8570) organism. Sequences with no matches were then searched against non-redundant protein sequences (nr) of the same organism. Proteins with the highest ranks in the BLASTP results were referenced to determine the functional annotations.

The transcripts from this study were named as “c” (component/contig), “g”’ (subgraph/trinity gene) and “i” (path sequence/trinity isoform) followed by numbers. The letter “F” stood for female, “M” for male, “PL” for partial-length and “FL” for full-length transcripts.

### 5.5. Multiple Sequences Alignment and Gene Tree Reconstruction

The most highly toxin transcripts (TPM values ≥ 100) were conceptually translated into protein sequences and subjected to alignment using Clustal Omega [82]. Alignments of the first-described and previously reported sequences in Russell’s viper are displayed in the manuscript and Appendix A, respectively. The conserved domains and functional motifs are identified and referenced in the figures. For some toxin gene families, the alignments were used to reconstruct in the program RAxML [83] using the PROTGAMMABLOSUM62 model with 1000 bootstrap replicates.

## Figures and Tables

**Figure 1 toxins-15-00309-f001:**
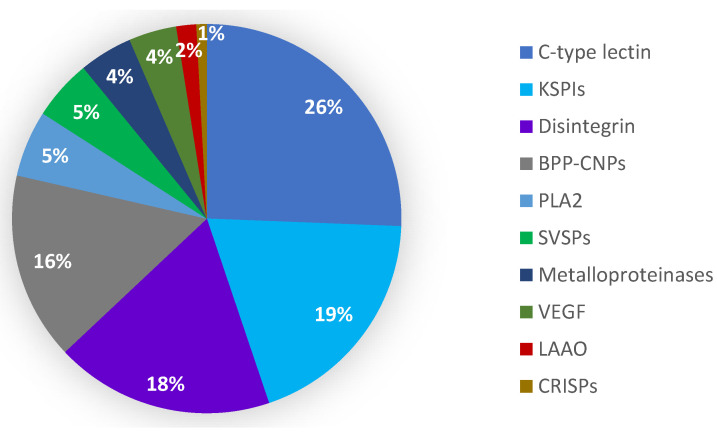
Overall toxin transcript (the top ten highly expressed toxin families) distribution in Myanmar Russell’s viper venom gland.

**Figure 2 toxins-15-00309-f002:**
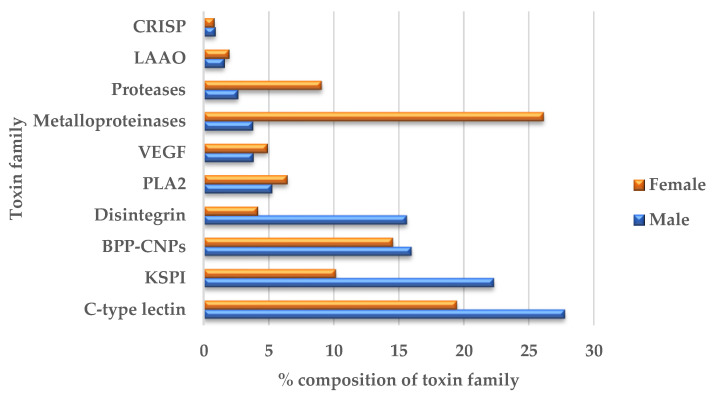
Comparison of toxin transcriptome compositions (the top ten highly expressed toxin families) of male and female snakes.

**Figure 3 toxins-15-00309-f003:**
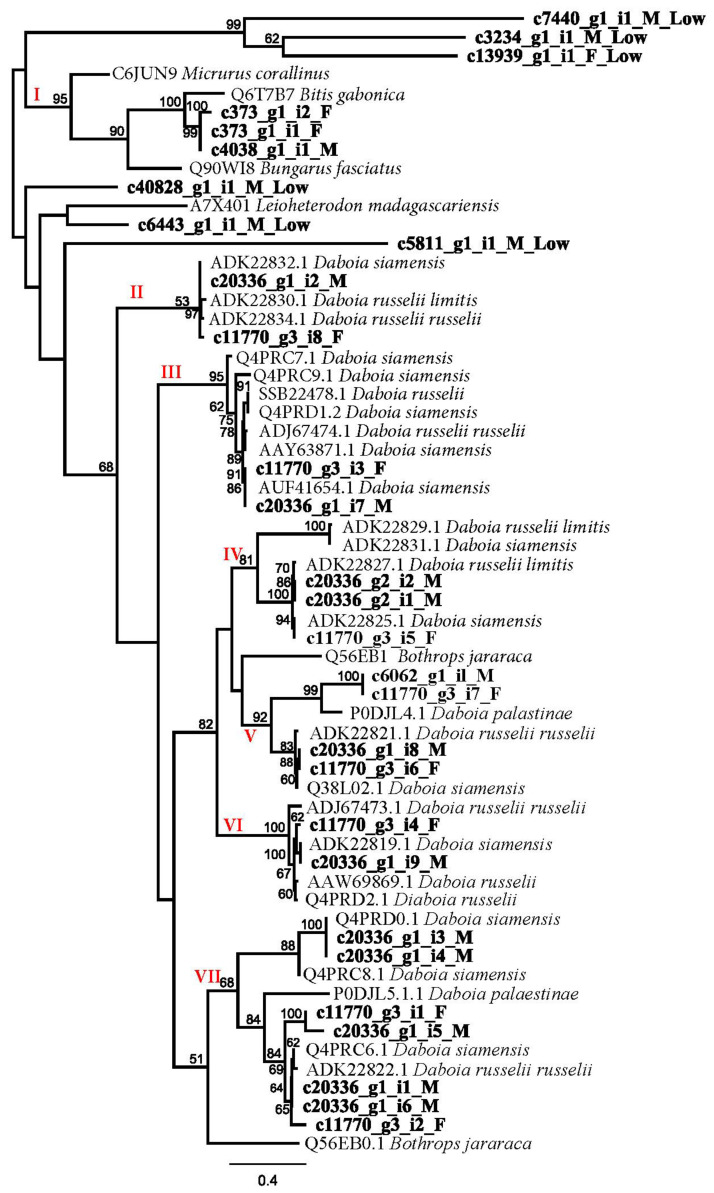
The maximum likelihood C-type lectins (CTLs) gene tree including *D. siamensis* with CTLs from other species. Bootstrap values greater than 50 are shown. Subtypes based on sequence alignments are indicated in Roman numerals. Refer to Appendix A for highly expressed transcript TPM values.

**Figure 4 toxins-15-00309-f004:**
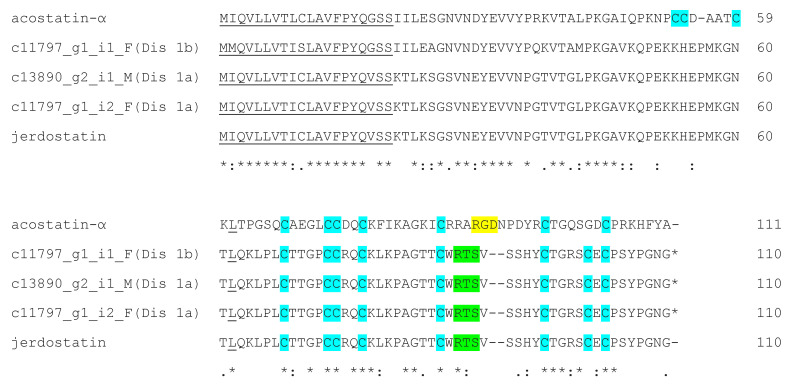
The peptide sequence alignment of Myanmar Russell’s viper disintegrin with acostatin-α from *D. siamensis* and jerdostatin from *Protobothrops jerdonii*. Signal peptides are underlined, while cysteine, RGD and RTS binding motifs are highlighted in blue, yellow and green correspondingly. (*) fully conserved residues; (:) strongly similar residues; (.) weakly similar residues.

**Figure 5 toxins-15-00309-f005:**
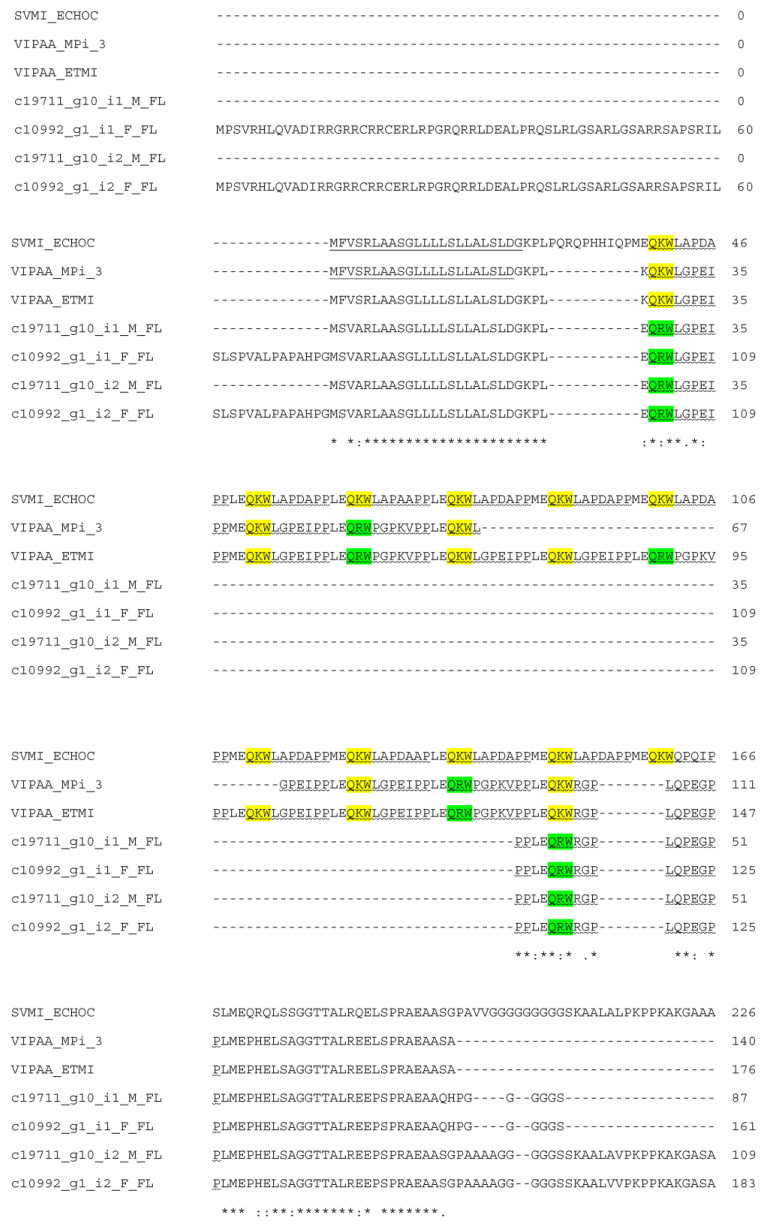
Sequence comparison of BPP-CNP precursors from Myanmar Russell’s viper with those of snake venom metalloprotease inhibitor (A8YPR6) from *Echis ocellatus* (SVMI_ECHOC) and MPi-3 (A0A6B7FNM6) (VIPAA_MPi-3) and endogenous tripeptide metalloproteinase inhibitor (A0A1I9KNP8) from *Vipera ammodytes ammodytes* aligned by Clustal Omega. Signal peptides are underlined. BPPs are shown by wavy underlines. C-type natriuretic peptides are highlighted in gray. Metalloproteinase tripeptide inhibitors, QKW and QRW are highlighted in yellow and green, respectively. (*) fully conserved residues; (:) strongly similar residues; (.) weakly similar residues.

**Figure 6 toxins-15-00309-f006:**
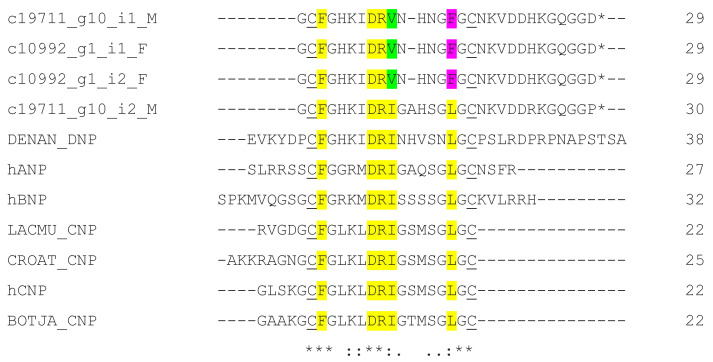
Sequence comparison of human natriuretic peptides (hANP (P01160), hBNP(P16860) and hCNP (P23582)) with those of different snake venoms (DENAN_DNP (P28374) from *Dentroaspis angusticeps*, LACMU_CNP (Q27J49) from *Lachesis muta muta*, CROAT_CNP (P0CV87) from *Crotalus atrox* and BOTJA_CNP(Q6LEM5) from *Bothrops jararaca*) aligned by Clustal Omega. Conserved cysteine residues are underlined, while relatively conserved amino acid residues known for NPR binding are highlighted. (*) fully conserved residues; (:) strongly similar residues; (.) weakly similar residues. Important residues for NP receptor binding are highlighted in yellow color with variable residues are in green and pink color.

**Figure 7 toxins-15-00309-f007:**
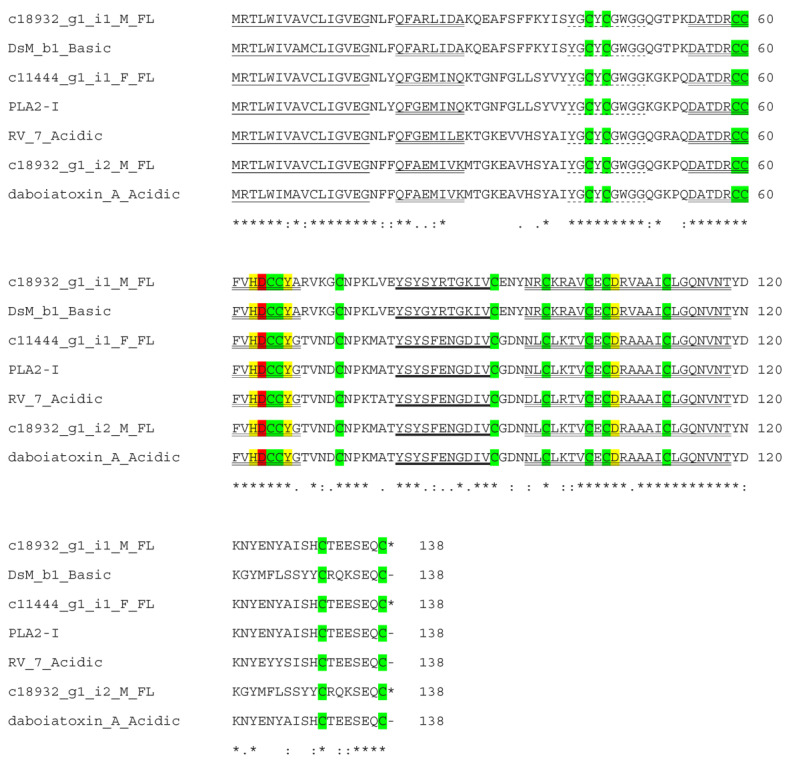
Alignment of deduced amino acid sequences of highly expressed full-length PLA_2_ transcripts from transcriptome with basic PLA_2_ DsM-b1 (A8CG82), PLA_2_-I (Q7ZZQ1), acidic PLA_2_ RV-7 (P31100) and daboiatoxin A (Q7T2R1) from *D. siamensis*. Signal peptide is single-underlined. Catalytic His48, Tyr52 and Asp90 are highlighted in yellow. Ca^2+^ binding motif Y25-GCYCGX-GG33 is dotted-underlined. α-helices are double-underlined. The region of β-strands of the β-wing are bold-underlined. The conserved Aspartate residues at 49 position are shown in red. Cysteine residues for disulfide bonds are in green. (*) fully conserved residues; (:) strongly similar residues; (.) weakly similar residues.

**Figure 8 toxins-15-00309-f008:**
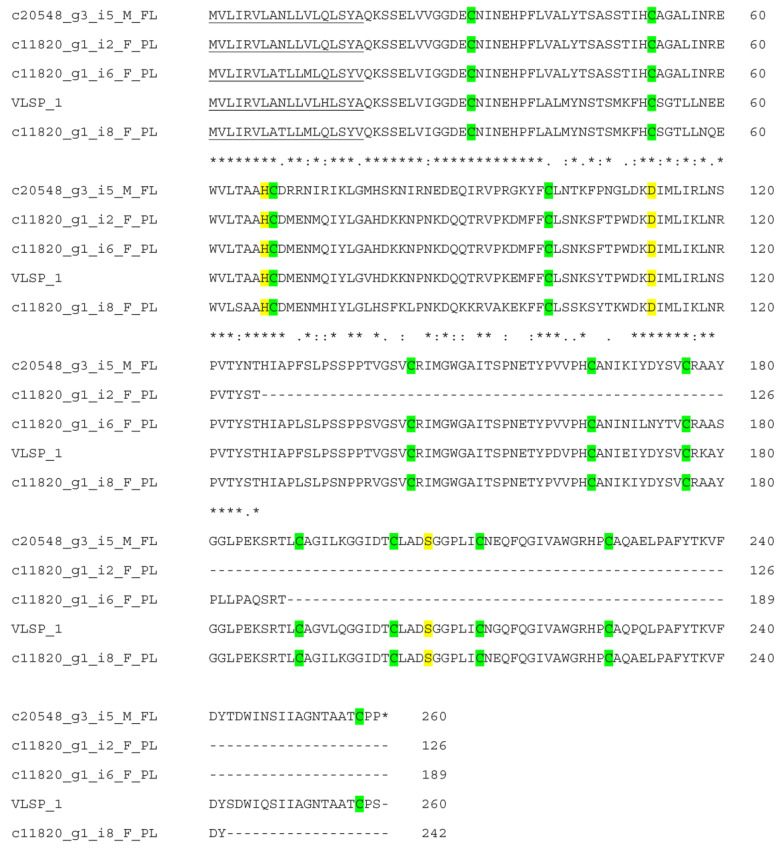
Sequence alignment of VLSP-1 from *Macrovipera lebetinus* and deduced amino acid sequences of transcripts from the transcriptomes. Signal peptide is single-underlined. Catalytic triad His57, Asp102 and Ser195 are highlighted in yellow. Cysteine residues in six disulfide bridges are highlighted in green. (*) fully conserved residues; (:) strongly similar residues; (.) weakly similar residues.

**Figure 9 toxins-15-00309-f009:**
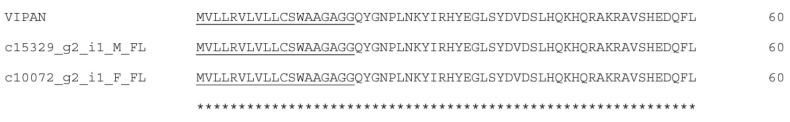
Sequence alignment of VIPAN, A disintegrin and metalloproteinase domain (ADAM)-containing protein 10 from *Vipera anatolica senliki* (A0A6G5ZVR9) and deduced amino acid sequences of c15329_g2_i1_M_FL and c10072_g2_i1_F_FL. Signal peptide is single-underlined. Zinc dependent metalloproteinase domain is double-underlined. Disintegrin domain is dotted-underlined. Cysteine residues are highlighted in green. Zinc binding site and RCD sequence are highlighted in yellow. (*) fully conserved residues; (:) strongly similar residues.

**Figure 10 toxins-15-00309-f010:**
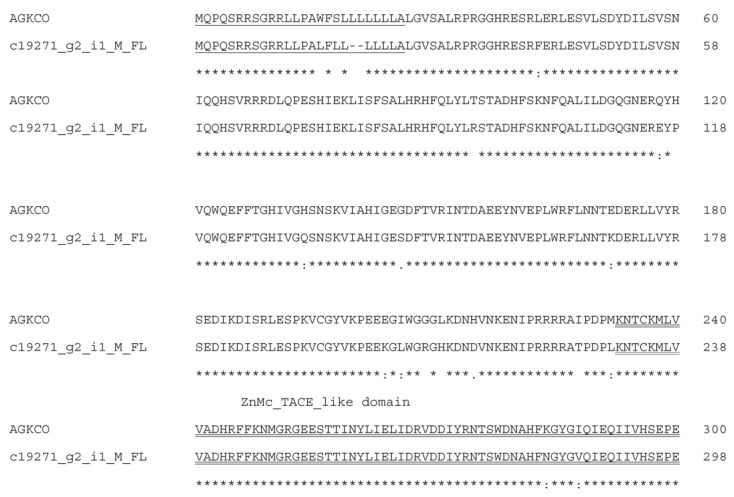
Sequence alignment of AGKCO (snake venom metalloproteinase from *Agkistrodon contortrix contortrix*, A0A1W7RJN7) and deduced amino acid sequences of c19271_g2_i1_M_FL. Signal peptide is single-underlined. Zinc dependent metalloproteinase domain is double-underlined. The disintegrin domain is dash-underlined. ADAM17-MPD domain is wave-underlined. CxxC motif (membrane-proximal domain) is highlighted in red. PxxP motif is highlighted in pink. Zinc binding site and EECD sequence are highlighted in yellow. Cysteine residues are highlighted in green. (*) fully conserved residues; (:) strongly similar residues; (.) weakly similar residues.

**Figure 11 toxins-15-00309-f011:**
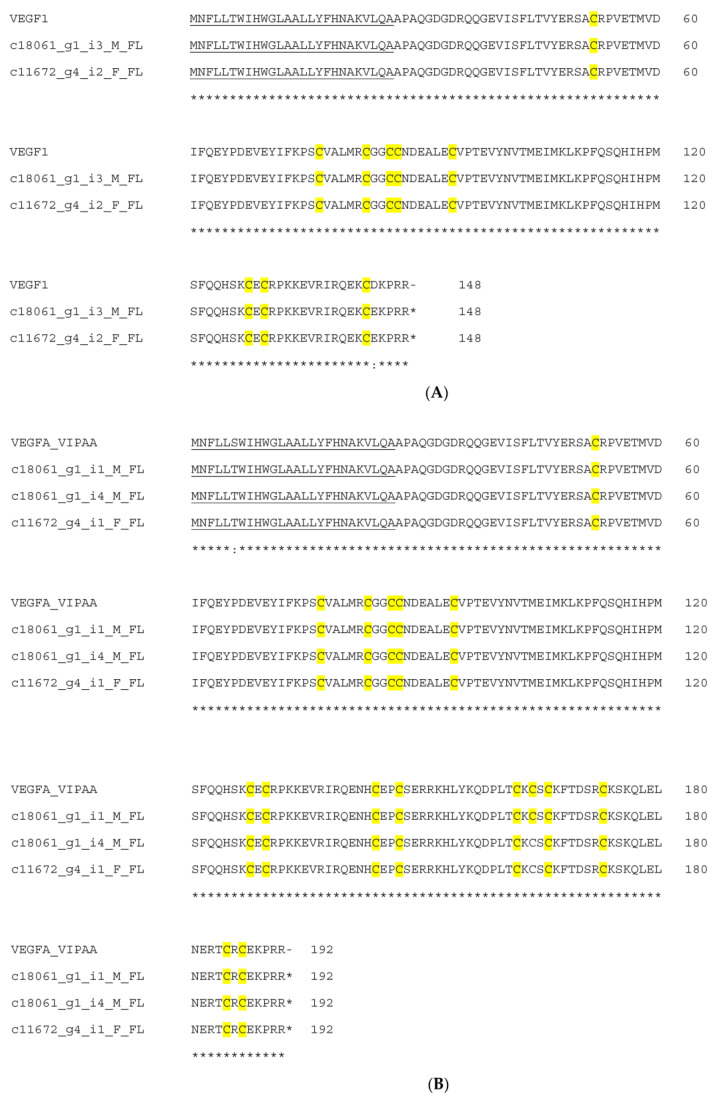
Sequence alignment of deduced amino acid sequences of VEGF transcripts with vascular endothelial growth factor 1 (VEGF1) from *Vipera anatolica senliki* (QHR82857) (**A**) and vascular endothelial growth factor A precursor (VEGFA-VIPAA) from *Vipera ammodytes ammodytes* (C0K3N5) (**B**). Signal peptide is single-underlined. Cysteine residues are in yellow color. (*) fully conserved residues; (:) strongly similar residues.

**Table 1 toxins-15-00309-t001:** The sequencing and the assembly quality of the venom gland transcriptomes of Myanmar Russell’s vipers (*Daboia siamensis*).

Trinity Assembly Parameters	Male	Female
Total raw reads	82,251,232	79,578,046
Total clean reads	78,647,670	74,444,200
Total clean nucleotides (nt)	7,752,565,867	7,235,655,822
Q20 percentage	98.93%	98.55%
GC percentage	46.26%	53.43%
Contigs created	88,325	50,858
Total assembled bases	75,163,976	39,200,694
Mean length (nt)	850.99	770.79
N50	1870	1379
Unigenes/transcripts assembled (only longest isoform per gene)	77,668	46,884

**Table 2 toxins-15-00309-t002:** Kunitz-type protease inhibitor (KSPI) transcripts from Myanmar Russell’s viper transcriptomes.

No.	Contig ID	Annotation	Accession No.	Species	Amino Acid	TPM Value
1	c17840_g2_i1_M_FL	Kunitz-type serine protease inhibitor 1	Q2ES50	*Daboia russelii*	84	109,755.22
	c11654_g1_i1_F_FL					16,745.60
2	c6152_g1_i1_M_FL	Serine protease inhibitor 1	A0A6G5ZVV7	*Vipera anatolica senliki*	148	1.59
	c6152_g1_i2_M_FL					0.76
3	c3688_g1_i1_M_FL	Serine protease inhibitor 4	A0A6G5ZUI5	*Vipera anatolica senliki*	514	26.46
	c5062_g2_i1_F_FL					9.09
4	c12297_g1_i1_M_FL	Tissue factor pathway inhibitor	U3TBN9	*Protobothrops flavoviridis*	303	2.45
5	c1956_g1_i1_M_PL	Serine protease inhibitor 3	A0A6G5ZW53	*Vipera anatolica senliki*	103	1.48
6	c12495_g1_i1_M_PL	Serine protease inhibitor 6	A0A6G5ZUI2	*Vipera anatolica senliki*	272	56.61
	c12495_g1_i2_M_PL					50.11
	c12495_g1_i3_M_PL					74.49
	c2247_g1_i2_F_PL					51.7
	c2247_g1_i1_F_PL					33.08
7	c15981_g1_i1_M_PL	kunitz/BPTI-like toxin	A0A6P9AX37	*Protobothrops guttatus*	139	47.25
8	c11558_g3_i1_F_PL	kunitz/BPTI-like toxin	XP_015682430.1	*Protobothrops mucrosquamatus*	139	25.45

**Table 3 toxins-15-00309-t003:** Transcripts with snake venom serine protease (SVSP) annotation.

No.	Contig ID	Annotation	Accession No.	Species	Amino Acid	TPM Value
1	c20548_g3_i4_M_FL	Factor V activator RVV-Vγ	P18965	*Daboia siamensis*	259	5495.65
2	c11820_g1_i3_F_PL	Thrombin-like_enzyme_TLBm (Factor V activator RVV-Vγ)	P18965	*D. siamensis*	251	2205.37
3	c11820_g1_i1_F_PL	Thrombin-like_enzyme_TLBm (Factor V activator RVV-Vγ)	P18965	*D. siamensis*	241	1794.30
4	c11820_g1_i4_F_PL	Thrombin-like_enzyme_TLBm (Factor V activator RVV-Vγ)	P18965	*D. siamensis*	251	1355.44
5	c38983_g1_i1_F_PL	α-fibrinogenase-like	E5L0E3	*D. siamensis*	105	3768.67
6	c20548_g3_i1_M_PL	α-fibrinogenase-like	E5L0E3	*D. siamensis*	131	3196.39
7	c20548_g3_i3_M_PL	α-fibrinogenase-like	E5L0E3	*D. siamensis*	155	952.41
8	c11820_g1_i9_F_PL	Thrombin-like_enzyme_catroxobin-1 (β-fibrinogenase-like)	E5L0E4	*D. siamensis*	126	981.63
9	c20548_g3_i5_M_FL	Serine protease VLSP-1	E0Y418	*Microvipera lebetina*	265	3027.55
10	c11820_g1_i8_F_PL	Thrombin-like_enzyme_TLBm (Serine protease VLSP-1)	E0Y418	*M. lebetina*	242	1400.97
11	c11820_g1_i6_F_PL	Thrombin-like_enzyme_TLBm (Serine protease VLSP-1)	E0Y418	*M. lebetina*	189	816.73
12	c11820_g1_i2_F_PL	Thrombin-like_enzyme_TLBm (Serine protease VLSP-1)	E0Y418	*M. lebetina*	126	448.44
13	c11820_g1_i5_F_PL	Thrombin-like_enzyme_TLBm (Serine protease VLSP-3)	E0Y420	*M. lebetina*	134	1318.30
14	c11820_g1_i7_F_PL	Thrombin-like_enzyme_catroxobin-1 (β-fibrinogenase-like)	E5L0E4	*D. siamensis*	104	919.53

## Data Availability

The raw data of the transcriptome assembly were uploaded to NCBI database under BioProject PRJNA545823, Biosample SAMN11938797 for males and SAMN11939170 for females.

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
