# Peer review of "Exploring Toxin Genes of Myanmar Russell’s Viper, Daboia siamensis, through De Novo Venom Gland Transcriptomics"

_toxins, 2023, doi:10.3390/toxins15050309_

Round 1

Reviewer 1 Report

The manuscript describes the transcriptome analyses of venom glands of Myanmer Russell’s viper, Daboia siamensis and reports 23 toxin genes families including 53 full-length transcripts. The purpose is clear. The results are well described, interesting and important in the field. However, I noticed a few concerns described below. Therefore, I recommend minor revision of the manuscript for the acceptance.

1.    Some terms used in figures are not clearly explained in legends, such as VEGFA-VIPAA in Figure 9 and Dr_CRPK in Supplementary Figure S6.

“VIPAN” in Figure 7 and “AGKCO” in Figure 8 are also better to be mentioned in the main text.

2.    Some paragraphs (for example the last two paragraphs) in Discussion are not directly related to the findings of the manuscript and unnecessary.

3.     D. russelli” in L573 is a mis-spelling of “D. russelii” (one “l” and two “i”s) (A very common spelling confusion).

Author Response

  1. Some terms used in figures are not clearly explained in legends, such as VEGFA-VIPAA in Figure 9 and Dr_CRPK in Supplementary Figure S6.

“VIPAN” in Figure 7 and “AGKCO” in Figure 8 are also better to be mentioned in the main text.

They are now explained in figure and added in the paragraph. Thank you for noting this.

  1. Some paragraphs (for example the last two paragraphs) in Discussion are not directly related to the findings of the manuscript and unnecessary.

These paragraphs were retained in the initial submission to link some of the identified toxin candidates with envenoming symptoms. After considering these comments and the subsequent revision necessary to streamline this manuscript last two paragraphs have been removed.

  1. D. russelli” in L573 is a mis-spelling of “D. russelii” (one “l” and two “i”s) (A very common spelling confusion).

It is corrected. Thank you.

Reviewer 2 Report

- Congratulations for the manuscript. It presents many novelties when compared with the manuscript by Yee and Rojnuckarin, 2020.

- Why didn't the authors show a figure for the phospholipases A2 alignment?

- Line 42: "Snake venoms are complex peptide cocktails containing hundreds of different molecules." This is a common misconception. In fact, many venoms, including many snake venoms are relatively simple mixtures (compared to for example a cell lysate). 

- Line 55-57: I don't understand if this paragraph refers to snake venoms or just the D. siamensis species?

- Line 85: There are 23 toxin gene families. Figure 1 shows 10 families. What are the 13 families not shown in the figure?

- Line 106: Correct: "... while PLA2, VEGF, Metalloproteinases, serine proteases and LAAO."

- Line 173: Correct: acostatin-α 

- Line 184: Correct: acostatin-α 

- Line 212: Correct: Bothrops jararaca

- Line 424: Correct: " .... in Figure 9 A and B."

- Line 517: " The snake venom disintegrins are a group of small cysteine-rich proteins found in all snake families." I think disintegrins are not found in all snake families.

- Line 570: Correct: 25 kDa

- Line 606: toxicogenomic or toxinogenomic?

- Line 657-672: Correct: font size. 

Author Response

- Congratulations for the manuscript. It presents many novelties when compared with the manuscript by Yee and Rojnuckarin, 2020.

Thank you very much. The authors appreciate this.

- Why didn't the authors show a figure for the phospholipases A2 alignment?

The PLA2 alignment (Figure 7) was constructed using the same approach and associated text was added to the manuscript to describe this.

- Line 42: "Snake venoms are complex peptide cocktails containing hundreds of different molecules." This is a common misconception. In fact, many venoms, including many snake venoms are relatively simple mixtures (compared to for example a cell lysate). 

It is now changed to “Snake venoms are mixtures of proteins working synergistically to produce various toxic functions”.

- Line 55-57: I don't understand if this paragraph refers to snake venoms or just the D. siamensis species?

The paragraph is deleted.

- Line 85: There are 23 toxin gene families. Figure 1 shows 10 families. What are the 13 families not shown in the figure?

Only highly expressed 10 toxin families are showed in Fig.1 for space limitation and clear labelling.

- Line 106: Correct: "... while PLA2, VEGF, Metalloproteinases, serine proteases and LAAO."

It is Corrected according to top highly expressed toxin families in male and female transcriptomes.

- Line 173: Correct: acostatin-α 

It is corrected.

- Line 184: Correct: acostatin-α 

It is corrected.

- Line 212: Correct: Bothrops jararaca

It is corrected.

- Line 424: Correct: " .... in Figure 9 A and B."          

It is corrected.

- Line 517: " The snake venom disintegrins are a group of small cysteine-rich proteins found in all snake families." I think disintegrins are not found in all snake families.

It is corrected.

- Line 570: Correct: 25 kDa

It is corrected.

- Line 606: toxicogenomic or toxinogenomic?

This is Toxicogenomic, in reference to tissue specific symptoms response to venom proteins post envenomation.

- Line 657-672: Correct: font size. 

It is corrected.

Reviewer 3 Report

The manuscript decribe the venom composition of Daboia siamensis using venom gland transcriptomic data.

Despite the high relevance of the study describing the venomics of the species, the authors did not compare their data to a previous study that performed a similar analysis in Dabaio siamensis, which also used proteomic and genomic approaches. Moreover, there is major concerns related to the methodology applied in the analysis, which may affect the results obtained and discussed in the manuscript (as listed below). After addressing these major concerns and proper modification performed, the manuscript can be re-analyzed to be considered for publication in Toxins.

Major review:

The authors only use one tool to perform de novo transcriptome assembly (i.e., Trinity) using the RNA-seq data. It was shown that more than one assembler is necessary to recover most toxin transcripts and ensure a complete picture of toxin components (Holding et al., 2018).

Integrating the assembly of more than one approach (such as rnaSPAdes, Bridger, NGEN, velvet, etc) with certainly recover more full-length toxin CDSs. It may modify the final toxin identification description and also the final toxin expression level of each toxin family.

The toxin gene annotation and identification rely on the use of Venomix. This tool was shown to recover a lower number of true toxins and also identify longer CDSs (Nachtigall et al., 2021), which represent toxins with erroneous start codon assignment. There are other alternatives available, like ToxCodAn, that allow retrieving a more confident toxin annotation, which may help to better characterize the venom composition of the species analyzed.

The authors did not mention applying any method to check for chimeric transcripts, which is a common feature when assembling toxin transcripts (Holding et al., 2018). It is an issue that must be carefully addressed when analyzing the de novo transcriptome assembly of venom glands to ensure a high-quality final toxin set.

The authors did not mention the mismatch rate considered when mapping reads and estimating expression levels using RSEM. By default, RSEM considers a mismatch rate of 10% (i.e., from 0 to 10% of mismatch of the mapped reads and transcripts is considered to estimate the expression level). Considering the evolution of multi-loci toxins, like CTLs, SVMPs, SVSPs, and PLA2s, it is important to consider a lower mismatch rate (e.g., 2%) to only account for the allelic variations. The modification of the mismatch parameter may result in differences in the final venom composition observed.

The authors could take advantage of running phylogenetic approaches to reveal the similarity of all toxin isoforms to other known viper toxin sequences and show alignments in supplementary files. It will certainly bring fruitful discussions related to toxin evolution in the genus/species.

The authors must include the data published for Daboi siamensis (Saethang et al., 2022) and also for other Daboia species to perform a comparative analysis. It will certainly improve the final discussion and bring novel insights related to Dabio venom evolution.

References:

Holding, M. L., Margres, M. J., Mason, A. J., Parkinson, C. L., & Rokyta, D. R. (2018) Evaluating the performance of de novo assembly methods for venom-gland transcriptomics. Toxins, 10(6), 249.

Nachtigall, P. G., Rautsaw, R. M., Ellsworth, S. A., Mason, A. J., Rokyta, D. R., Parkinson, C. L., & Junqueira-de-Azevedo, I. L. (2021) ToxCodAn: a new toxin annotator and guide to venom gland transcriptomics. Briefings in Bioinformatics, 22(5), bbab095.

Saethang, T., Somparn, P., Payungporn, S. et al. (2022) Identification of Daboia siamensis venome using integrated multi-omics data. Sci Rep 12, 13140.

Author Response

The authors only use one tool to perform de novo transcriptome assembly (i.e., Trinity) using the RNA-seq data. It was shown that more than one assembler is necessary to recover most toxin transcripts and ensure a complete picture of toxin components (Holding et al., 2018).

Integrating the assembly of more than one approach (such as rnaSPAdes, Bridger, NGEN, velvet, etc) with certainly recover more full-length toxin CDSs. It may modify the final toxin identification description and also the final toxin expression level of each toxin family.

We agree with the reviewer and will take this comment in consideration in the future. However, computational resources available to researchers in Myanmar are limited. Although it is our desire to explore these alternative approaches more thoroughly, we were limited to certain computing capabilities. Beyond Trinity, we also used CLC Genomic Workbench for assembly and comparing the two assembly statistics (Supplementary Table S4). The table and some statements regarding these results were added to the manuscript in the discussion section. The assembly results from Trinity are more consistent. Therefore, we used its output in further steps in the Venomix pipeline.

The toxin gene annotation and identification rely on the use of Venomix. This tool was shown to recover a lower number of true toxins and also identify longer CDSs (Nachtigall et al., 2021), which represent toxins with erroneous start codon assignment. There are other alternatives available, like ToxCodAn, that allow retrieving a more confident toxin annotation, which may help to better characterize the venom composition of the species analyzed.

Venomix was not used to be a toxin ‘identifier’ or ‘predictor’. It organizes the outputs of common programs used in venom transcriptomic analysis Like in ToxCodAn, candidate toxins are identified in Venomix platform via BLAST, with the initial search done outside of the main program.

We attempted to use ToxCodAn. However, the search program has never ran to a completion. After troubleshooting the errors, the CodAn.py script failed to return any toxin predictions despite the built-in BLAST search identifying 221 hits for the male transcriptome and 175 hits for the female transcriptome. This problem has not been resolved despite our multiple contacts to ToxCodAn team, and it is not clear whether the models simply did not match our toxins or there were other issues.

In regard to a very understandable suggestion to explore the dataset by means of several assembly or annotation pipelines, we can comment that this study was focused on the most highly abundant transcripts from recently milked snake venom glands. This ensured a selection of genes that were highly expressed shortly after milking, allowing us to consider these to be true toxins in high abundance. The venom cocktail is replenished with high rates of transcription, and we also reserved samples for the proteomic analysis to confirm our main findings. This is just the first gene expression study conducted on this poorly studied snake species, and we hope to explore ToxCodAn and similar annotation programs in the future. We acknowledge that there are limitations to our approach and have addressed this early on in the discussion section.

The authors did not mention applying any method to check for chimeric transcripts, which is a common feature when assembling toxin transcripts (Holding et al., 2018). It is an issue that must be carefully addressed when analyzing the de novo transcriptome assembly of venom glands to ensure a high-quality final toxin set.

Chimeric transcripts are possible, and we hope to address this aspect in a future investigation when we are able to complement our results with proteomic and genomic data. For now, we were able to construct high-quality Trinity assemblies from which the final toxin set was derived.

The authors did not mention the mismatch rate considered when mapping reads and estimating expression levels using RSEM. By default, RSEM considers a mismatch rate of 10% (i.e., from 0 to 10% of mismatch of the mapped reads and transcripts is considered to estimate the expression level). Considering the evolution of multi-loci toxins, like CTLs, SVMPs, SVSPs, and PLA2s, it is important to consider a lower mismatch rate (e.g., 2%) to only account for the allelic variations. The modification of the mismatch parameter may result in differences in the final venom composition observed.

Our QUAST report indicated that there was a zero mismatch in the performance of the Trinity tools. Again, we hope to evaluate this in future studies of this species venom transcriptome, when we can compile a more comprehensive transcriptomic and proteomic dataset to provide more robust analysis and to evaluate alternative assembly approaches.

The authors could take advantage of running phylogenetic approaches to reveal the similarity of all toxin isoforms to other known viper toxin sequences and show alignments in supplementary files. It will certainly bring fruitful discussions related to toxin evolution in the genus/species.

Thank you very much for your valuable suggestion. Phylogenetic analyses of all the toxins are in our plans as we expand this work further.

The authors must include the data published for Daboi siamensis (Saethang et al., 2022) and also for other Daboia species to perform a comparative analysis. It will certainly improve the final discussion and bring novel insights related to Dabio venom evolution.

Thank you. The paper of Saethang et al (2022) described a proteomic study, and we cited it in the introduction section. It is also a useful resource for planning our proteomics analysis

 References:

Holding, M. L., Margres, M. J., Mason, A. J., Parkinson, C. L., & Rokyta, D. R. (2018) Evaluating the performance of de novo assembly methods for venom-gland transcriptomics. Toxins, 10(6), 249.

 Nachtigall, P. G., Rautsaw, R. M., Ellsworth, S. A., Mason, A. J., Rokyta, D. R., Parkinson, C. L., & Junqueira-de-Azevedo, I. L. (2021) ToxCodAn: a new toxin annotator and guide to venom gland transcriptomics. Briefings in Bioinformatics, 22(5), bbab095.

Saethang, T., Somparn, P., Payungporn, S. et al. (2022) Identification of Daboia siamensis venome using integrated multi-omics data. Sci Rep 12, 13140.

Thank you very much for the suggested references, we have cited them in our manuscript.

Reviewer 4 Report

The manuscript under review reports the transcriptomic analysis of the venom gland of the snake D. siamensis from Myanmar. The subject itself is interesting given the limited literature on the topic and the presence of some very interesting toxins reported in previous work on the same venom (Yee et al. 2018). The methodology used is appropriate for obtaining a vast bank of sequences, but the way in which this work is presented is not enough to advance the knowledge of the venom composition of this species. There are dubious interpretations of the literature, and the data are presented in a manner that does not help to understand their relevance. In addition, it is not clear why to present a redundancy of alignments using some of the sequences obtained, and not of others. The discussion is poor and does not add much to what is known about viper venoms, nor does it highlight points that could be interesting and peculiar to the venom of D. siamensis from Myanmar.

Specific points:

Line 55: In the statement “Two of the most abundant toxins, snake venom serine proteases (SVSPs) and venom phospholipase A2 (vPLA2) increased in evolutionary mutation rates since the divergence of viperidae, while snake venom 56 metalloproteinases (SVMPs) decreased [21]” is very odd and I`ve found no evidence of the decreased evolutionary mutation rates in SVMPs the literature or in the selected reference.

Line 58: In the statement: “Despite these variations, the Russell’s viper protein-coding toxin genes had not been investigated through a comprehensive transcriptomic approach.” That is correct, but there are previous reports about D. siamensis venom composition that should be cited in the introduction.

Concerning the characterization of the transcripts:

·       In the characterization of transcripts, did you consider coverage? Apparently, some transcripts are very short.

·       A list of the 53 full-length sequences with the toxin gene family, TPM counting in the venom gland and non-venom transcriptomes should be included.

Figure 1:

·       The chart of Figure 1 should include the 23 toxin families identified. Why only 10?

·       Names of toxin groups are misleading. In most papers, the group presented as “protease inhibitor” is usually referred to as Kunitz-type serine protease inhibitor;

·       SVMPI is generally used for Class I snake venom metalloproteinases. These transcripts include the tripeptide which inhibits SVMP, but they also contain NP and BPPs. In most papers these transcripts are classified as BPPs;

·       The disintegrins reported here are actually a particular class of toxins with disintegrin activity encoded by RPTLN genes. I suggest classifying them as RPTLN, avoiding confusion with the disintegrins generated from metalloproteinase genes.

Figure 2:

·       Did you consider all transcripts to make the total of each toxin gene family?

·       A Table similar to supplementary Table 1 should be included in the main text, containing only the 53 full-length transcripts with identification and expression levels (TPM), including the comparison of their TPM in male and female transcriptomes.

In the following Figures (3-9) how did you select the transcripts that should be included in the alignments?

Figure 3: If you have 8 subtypes of CTL, why show alignments of only one subtype? and only 3 sequences out of the 23? A much better representation would be the phylogenetic tree shown in supplementary material, including the transcripts and one named sequence representative of each subtype.

Table 2: I could not recognize the 6 contigs with full-length sequences. Worse than that, the transcript with the highest expression has very limited coverage.

"Disintegrins”:

·       Only two sequences are shown. What about the third sequence? Does it not contain RTS? Why not included it in the alignment?

·       The transcript c11797g1 is identical in male or female? Is this just the third sequence

Figure 5: The alignments should contain the full-length sequences, including the NP, BPP and SVMP inhibitor tripeptides. In my view, it makes no sense to show alignments of such a partial sequence

PLA2: It was not clear. It appears that you did not find any homologous sequences for the 5 full-length sequences. Please explain.

Supplementary Tables: What is the label yellow or pink? Please include %coverage of each transcript.

Author Response

Line 55: In the statement “Two of the most abundant toxins, snake venom serine proteases (SVSPs) and venom phospholipase A2 (vPLA2) increased in evolutionary mutation rates since the divergence of viperidae, while snake venom 56 metalloproteinases (SVMPs) decreased [21]” is very odd and I`ve found no evidence of the decreased evolutionary mutation rates in SVMPs the literature or in the selected reference.

We have deleted these sentences. Thank you.

Line 58: In the statement: “Despite these variations, the Russell’s viper protein-coding toxin genes had not been investigated through a comprehensive transcriptomic approach.” That is correct, but there are previous reports about D. siamensis venom composition that should be cited in the introduction.

We added more citations on proteomic studies of Russell’s vipers in the introduction. Thank you for highlighting this, as it greatly improves the manuscript.

Concerning the characterization of the transcripts:

  • In the characterization of transcripts, did you consider coverage? Apparently, some transcripts are very short.

We are begging an apology, it is not clear what the referee means by “short” in this comment. Venomix pipeline identified toxin candidates with the similar threshold values of 1e-20. Although that is not necessarily reflecting matching sequence length, we also checked for Query coverage (%).

  • A list of the 53 full-length sequences with the toxin gene family, TPM counting in the venom gland and non-venom transcriptomes should be included.

A list of the 53 full-length sequences with toxin gene families, annotations of all transcripts and their TPM values were provided. We mapped the raw reads from the “non-venom” transcriptome onto our male and female transcriptomes to determine these values. We did not consider those values relevant as we were focused only on the highly expressed toxins. In every instance where venom transcript expression was greater than 100 tpm, the TPM values in the venom gland were (at the lowest) 150-fold higher in the venom gland than in a non-venom tissue. We did not consider include the non-venom gland results, however, we have added a comment to the results section to clarify this issue. Readers can find comprehensive TPM information from the Supplemental Table 2.

Figure 1:

  • The chart of Figure 1 should include the 23 toxin families identified. Why only 10?

Only highly expressed 10 toxin families are showed in Fig.1 due to the space limitation and to allow clear labelling.

  • Names of toxin groups are misleading. In most papers, the group presented as “protease inhibitor” is usually referred to as Kunitz-type serine protease inhibitor;

Thank you. The name of protease inhibitor is changed to KSPIs in Figure 1 and in the title of section 2.4.

  • SVMPI is generally used for Class I snake venom metalloproteinases. These transcripts include the tripeptide which inhibits SVMP, but they also contain NP and BPPs. In most papers these transcripts are classified as BPPs;

Thank you ‘SVMPI’ are changed to BPP-CNPs. We have also edited the text accordingly.

  • The disintegrins reported here are actually a particular class of toxins with disintegrin activity encoded by RPTLN genes. I suggest classifying them as RPTLN, avoiding confusion with the disintegrins generated from metalloproteinase genes.

Thank you. We changed the title and have edited the text accordingly.

Figure 2:

  • Did you consider all transcripts to make the total of each toxin gene family?

Yes, we considered all transcripts for each toxin gene family. We only described the highly-expressed 10 toxin families in Fig 1 and 2 to avoid overcrowded labeling.

  • A Table similar to supplementary Table 1 should be included in the main text, containing only the 53 full-length transcripts with identification and expression levels (TPM), including the comparison of their TPM in male and female transcriptomes.

We provide a supplementary table to illustrate it. We differentiate the transcripts for male (M) and female (F) samples in transcript IDs.

In the following Figures (3-9) how did you select the transcripts that should be included in the alignments?

We selected full-length and newly-discovered sequences to include in the alignments. 

Figure 3: If you have 8 subtypes of CTL, why show alignments of only one subtype? and only 3 sequences out of the 23? A much better representation would be the phylogenetic tree shown in supplementary material, including the transcripts and one named sequence representative of each subtype.

Thank you. We changed Supplementary Table S3 in Figure 3 and Supplementary Figure S1.

Table 2: I could not recognize the 6 contigs with full-length sequences. Worse than that, the transcript with the highest expression has very limited coverage.

Their Query coverage was 93-99%. In the amino acid column, values correspond to the actual amino acid lengths of the toxins.

"Disintegrins”:

  • Only two sequences are shown. What about the third sequence? Does it not contain RTS? Why not included it in the alignment?

Three sequences, c11797_g1_i1_F, c11797_g1_i2_F and c13890_g2_i1_M. Two isoforms were identified. All sequences contained RTS motif.

  • The transcript c11797g1 is identical in male or female? Is this just the third sequence

The c11797_g1_i1 and c11797g1_i2 sequences are from female.

Figure 5: The alignments should contain the full-length sequences, including the NP, BPP and SVMP inhibitor tripeptides. In my view, it makes no sense to show alignments of such a partial sequence

We have shown the mentioned alignments in our previous paper Yee, KT et al. (2017). Therefore, we omit that alignment in current manuscript and showed only details of the CNP sequences.

PLA2: It was not clear. It appears that you did not find any homologous sequences for the 5 full-length sequences. Please explain.

Thank you. We added the alignment (Figure 7) and an additional explanation in the text.

Supplementary Tables: What is the label yellow or pink? Please include %coverage of each transcript.

The colors were added to aid author annotation and analysis. They have been removed. The % coverage of each transcript comment is not clear. We included Isoform Principal Component (IsoPC) values, as these may be what the reviewer is requesting. As multiple BLAST queries matched to the same transcripts, we would have multiple %coverages to display.

Round 2

Reviewer 4 Report

Dear Editor

I checked the answer to my comments and apparently, the authors accepted part of the suggestions while they did not quite understand other comments. Unfortunately, I did not have access to the comments of the other reviewers that encouraged the acceptance of the manuscript. Most of all, I will not be able to carefully revise the new version of the manuscript in the extremely short time that you recommend. Considering the points above, I will mark accepted” but I suggest you follow the recommendations of the other reviewers.